# 'Age specific variations in ovarian reserves in healthy fertile and infertile women: A cross sectional study

Neena Malhotra[1]*, Pankush Gupta[2], Saloni Kamboj[1], Pradeep Chaturvedi[3], Rintu Kutum[4]

1 Department of Obstetrics and Gynecology, All India Institute of Medical Sciences Delhi, Delhi, India, 2 Sitaram Bhartia Institute of Science and Research, Delhi, India, 3 Department of Reproductive Biology, AIIMS, Delhi, India, 4 Ashoka University, Haryana, India

* malhotraneena@yahoo.com

**Data Availability Statement:** ll relevant data are within the manuscript and its Supporting Information files.

## Abstract

Ovarian reserve tests are valuable for evaluation of female fertility, and to formulate appropriate treatment strategies for infertile women. Antral follicle count (AFC) and Anti-Mullerian hormone (AMH) are most reliable markers of ovarian reserve which are related inversely to age. There are many factors that affect ovarian reserve like race, ethnicity, fertility status, BMI or any chronic illness. We conducted this study to find outage specific nomograms for AMH and AFC among fertile and Infertile Indian women, to find out any variations between fertile and Infertile ovarian reserves at various centiles, to define the age cut-off of decline in AMH and AFC among fertile and Infertile Indian women and to find correlation between AMH and AFC. It was a prospective cross sectional single centre study conducted at a tertiary hospital of northern India from March 2017 to February 2022. Fertile healthy women were recruited from family planning clinic, oocyte donors and subfertile women from Gynaecology and ART clinic. AMH was done using ELISA, Beckmann Coulter Gen II assay and AFC was done using TVS with high frequency probe (9.0 MHZ, Voluson,S-6, GE Healthcare, USA) by trained personnel. R Statistical Programming Language was used for statistical modelling and visualization. Age-specific AFC centile chart and AMH centile chart were generated using GAMLSS (Generalized Additive Models for Location Scale and Shape) package available in R Statistical Computing Language. A Non-linear decline in ovarian reserves among fertile, while linear among infertile women was seen. Centiles defined for both groups with a faster decline in infertile women. Age cut off for decline in AMH and AFC in fertile women approximately 31 years using ROC analysis and Age cut off for decline in AMH and AFC in infertile women is approximately 34 years. There seems to be a good correlation between AFC and AMH. We need to counsel women to consider child bearing well before ovarian reserves decline (31–34 years).

**Funding:** WHO is providing funding for publication assistance. REC-PDCL provided funds for study through CSR initiative.

**Competing interests:** The authors have declared that no competing interests exist.

## Introduction

Use of ART techniques have been on a rise with increase in infertility [1]. Over recent years, testing of ovarian reserve to predict future reproductive life has become crucial while women postpone childbearing globally [2]. Testing for ovarian reserve is fundamental in planning the course of infertility treatment and offering ART to those with borderline reserves besides predicting outcomes of treatment [3]. There are varieties of ovarian reserve tests that include ultrasound and biochemical parameters [3]. Antral follicle count (AFC) and anti-Mullerian hormone (AMH) have been shown to be the best markers of ovarian reserve [4, 5].

Anti-Mullerian hormone (AMH) is a dimeric glycoprotein which is a member of the transforming growth factor beta (TGF-b) superfamily, produced by granulosa cells of primordial follicles that have undergone initial recruitment, and is thought to reflect the size and quality of the ovarian reserve [6]. In addition, AMH inhibits follicular sensitivity to follicle stimulating hormone (FSH) and plays a role in the process of follicle selection and dominance. Currently, AMH is considered as an optimal biomarker that reflects ovarian reserve, convenient in detection and more effective than other biomarkers, including FSH and estradiol (E2). Measuring AMH during an initial fertility workup is crucial to plan ART cycles, including protocols, dosing of gonadotropins and also predicting the response to ovarian stimulation. These results can be used to determine the recombinant human FSH starting dose and predict the final oocyte yields and develop a nomogram that could predict oocyte yield. Lee et al. investigated the cutoff level of serum AMH for predicting poor (number of oocytes retrieved, ≤3), normal (4–19), and high responders (≥20). Especially for predicting poor responders, the cutoff level was 1.08 ng/mL, with 85.8% sensitivity and 78.6% sensitivity [7]. In a multicentric trial by Nelson et al, Anti-mullerian hormone proved to be a stronger predictor of ovarian response to gonadotropin therapy compared to AFC utilizing GnRH agonist and GnRH antagonist protocols. Antral follicle count provided no added predictive value beyond AMH [8]. AMH has also found usefulness in even predicting rate of euploid blastocyst besides live birth, after In-vitro fertilization/ intracytoplasmic cycles [9]. Further beyond infertility it is utilized by clinicians to assess the ovarian reserve and predict menopause age [10]. Therefore, the reference range of AMH in normal healthy women is needed to provide useful information on ovarian function that can be of potential clinical benefit. Anti-Mullerian hormone is measured using enzyme linked immunosorbent assays (ELISAs). Various assays are available: Gen II (Beckman Coulter); pico AMH (Ansh Labs); and Elecsys (Roche) [11]. Previous studies have mostly measured AMH levels using ELISAs, the Diagnostics Systems Laboratory (DSL 10–14400) assay or the Immunotech (IOT, A11893 IVD, EU) assay [12]. These assays utilize different primary antibodies against AMH and different standards, and consequently, the crude values reported differed substantially. With the consolidation of these 2 companies by Beckman Coulter in 2011, as sole ownership of the patent to measure mammalian AMH, there is finally a single commercially available assay: the AMH Gen II assay (A79765), which has fully replaced the DSL and IOT assays [13]. Therefore, using the Beckman Coulter Gen II assay to estimate the reference range for AMH is more practical. However, only a few studies established the reference range for AMH using the new Gen II assay, especially for Indian women [14].

Antral follicle count (AFC) also serves as an objective ovarian reserve test helping decide treatment protocols and anticipating response. AFC is the most commonly used ultrasound marker to identify ovarian reserve and is reliable and easy to measure [15]. It consists of counting of all follicles in the range of 2–10 mm visualized through high resolution transvaginal ultrasound examination. Assessment of AFC has been a simple method of predicting the occurrence of menopause and thus the duration of the reproductive lifespan [16]. It is well established that the female reproductive function deteriorates with age because of reduction of

the ovarian follicle pool [17]. Several studies of autopsy and surgical specimens showed that the number of antral follicles is related to the number of primordial follicles within the ovaries [18]. In particular, the number of small antral follicles decreases with age similarly to the number of primordial follicles [17, 19]. Indeed, several studies have demonstrated that AFC declines with chronologic age in women, although it is still not clear whether this decline has a biphasic [20] or a linear pattern [21, 22].

Ovarian reserve markers including AMH and AFC are prone to variations because of race, ethnicity, use of hormones including oral contraceptives [23]. Various studies have been published on nomograms based on AMH and AFC for fertile and infertile population [24, 25]. As the values are affected by race, ethnicity, it becomes prudent for each population to have a nomogram to be used as reference. Several AMH–age nomograms have been reported before; however, most of the studies used hospital inpatient samples, particularly samples from infertile women instead of healthy females. With improved access to ART services, more and more Indian women are seeking treatment and using American or European standards may not seem justifiable. There is also evidence that ovarian reserves among Indian women decline earlier compared to women from Europe [3, 26, 27]. However, this data comes from women who had undergone ART cycle and may not reflect the nomograms as would come from women other that the infertile cohort. The reference range for AMH levels obtained from a large, population-based sample of healthy women was believed to be more practical, reliable, and accurate in representing the normal population, therefore, additional studies were warranted to determine that.

Currently there are no age-related centile charts for AFC and AMH from Indian women. The present study was aimed to derive nomograms for AMH and AFC among fertile and infertile women of Indian origin, indicating the trends in decline and the relationship between two ovarian reserve markers. Additionally, we evaluated the relationship of AMH and AFC with age to provide an optimal model of decline in AMH and AFC with age.

## Materials and methods

In a cross-sectional study, conducted at a tertiary referral hospital in North India, both infertile and healthy fertile women who had borne one child conceived spontaneously were invited for testing ovarian reserves at one time. The study was approved by the ethics committee of the Institute (IEC-491/07.10.2016) and all participants gave informed written consents to be part of the study. The study was conducted from January 2017 until December 2022, with recruitment affected by the COVID pandemic, enrolling 3240 women; 1902 infertile and 1338 healthy women as controls. The participating women including infertile cohort attended fertility and ART clinics while fertile cohort consisted of those attending the family planning and contraception clinics or healthy oocyte donors with confirmed fertility.

The inclusion criteria for infertile women were age 21–40 years, with infertility of more than one year, while healthy age and BMI matched women were selected as controls. Women with conditions that could affect ovarian reserves including endometriosis, PCOS, previous ovarian surgery, immunological disorders and those on gonadotoxic therapy were excluded from both cohorts. Also, women with co-morbidities such as diabetes mellitus, morbid obesity were excluded. Informed consent was obtained from all women in both the cohorts, informing them that these tests will be confidential and not relevant on future child bearing especially in the fertile cohort.

Testing for both biomarkers of ovarian reserve was done early in the follicular phase between day 2 to day 6 of menstrual cycle ensuring no intake of oral contraceptive pills prior to testing in either group. Blood samples for AMH were centrifuged after collection were

aliquoted and stored at -80 degrees for future use within 4 weeks of blood collection. Serum Anti Mullerian Hormone was measured in all samples using the Beckman Coulter (M/S Immunotech) AMH Gen II ELISA kits. This kit uses the two-site 'sandwich' assay and has an analytical sensitivity of 0.08ng/ml. This method has got a validated correlation with the automated access AMH assay [28] The limit of detection for Gen II was 0.18 nmg/ml and the measurement range varied between 0.16 to 22.6 ngm/ml.

Antral follicle counts were done using high frequency transvaginal transducer (8.5 MHz; S-6, GE 3600 Healthcare, USA)when follicles between 2–8 mm were measures in three planes in each ovary, number totaled for both ovaries All scans were done the same personnel (NM, PG) who were trained sufficiently to ensure consistency. All data recorded on the proforma was entered in EXCEL and further statistical analysis was conducted using R software environment.

## Statistical analysis

Baseline characteristics such as age, BMI, AFC, AMH were measure for both the fertile as well as infertile cohorts, which were expressed as mean and standard deviations, percentages whichever was most appropriate. Both cohorts were divided into six sub groups based on age.

Age specific centile charts for AMH and AFC were generated as per the CG-LMS method using GAMLSS (Generalized Additive Models for Location Scale and Shape) package available in R Statistical Computing Language. This method produces a model that specifies the centiles in terms of age-specific curves called L (skewness), M (median) and S (coefficient of variation). At each age point, M curve corresponds to the median AFC/AMH level, S curve corresponds to the coefficient of variation and L curve allows for age dependent skewness of distribution of AMH/AFC. The value of any centile at a given age can be computed from the values of L,M and S. Six empirical centiles, including the 3rd, 10th, 25th, 50th, 75[th] and 90th centiles, and nomogram tables were constructed. Receiver Operator Characteristic(ROC) curve analysis was conducted to analyze the predictive accuracy of age with respect of AFC and AMH and to determine the cut-off values using the Area Under the Curve (AUC) analysis. The cut off values for AMH of <1.2 ngm/ml and AFC of < 5 was taken as per the POSEDION criteria in describing diminished ovarian reserves while generating the ROC [29].

## Results

Table 1 presents the baseline values of age, BMI, AMH and AFC values in the fertile and infertile groups.

**Table 1. Baseline parameters of fertile and infertile subjects.**

| Parameter | Infertile | Fertile |
|---|---|---|
| Number | 1902 | 1338 |
| Age* (Years±SD) | 31.1± 4.78 | 28.7±5.07 |
| Infertility | | |
| Primary, n (%) | 1486(78.2%) | - |
| Secondary, n (%) | 416(21.8%) | - |
| BMI* (Kg/m$^2$±SD) | 25.09±4.43 | 24.23± 2.23 |
| AMH* (ng/dl±SD) | 3.4± 2.48 | 4.37±2.62 |
| AFC*(n±SD) | 12±6.8 | 15±6.2 |

*- Mean, BMI- Body Mass Index, AMH- Antimullerian Hormone, AFC- Antral Follicle count

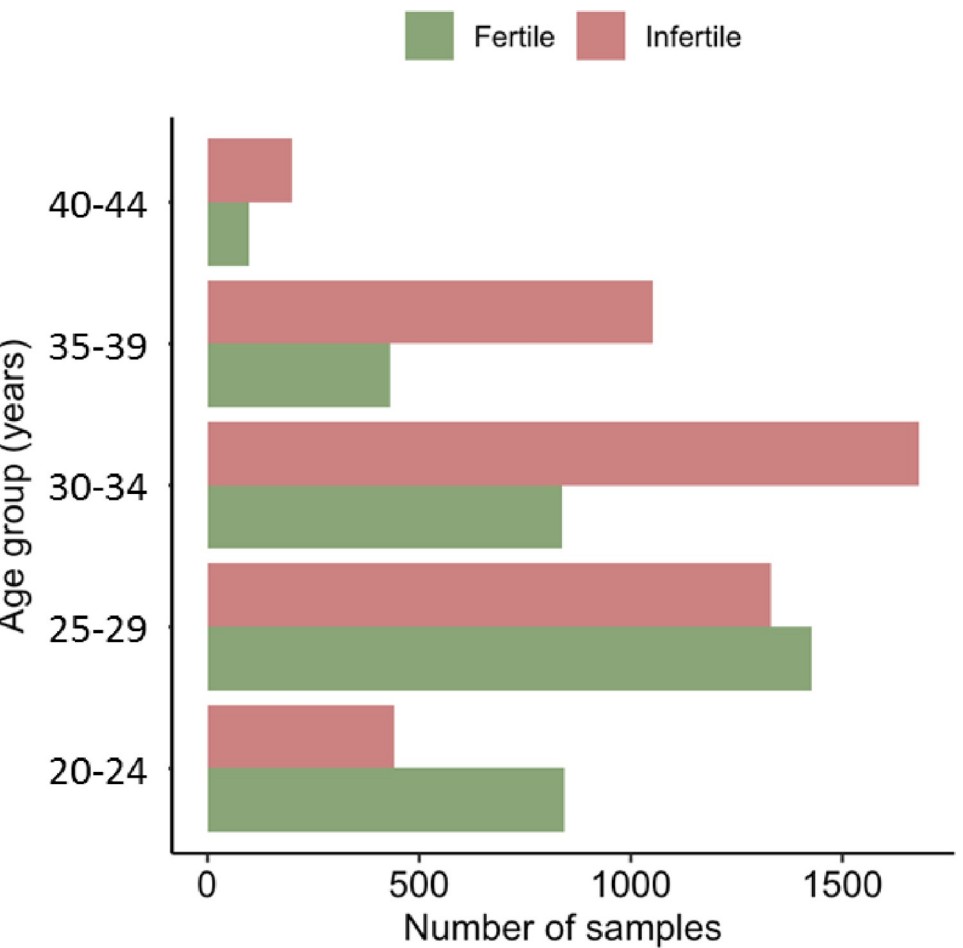

**Fig 1. Age segregated subgroup analysis of subjects.**

Age segregated number were comparable in the fertile and infertile groups across different sub groups, with lower number of women between ages 40 and beyond (Table 1 and Fig 1).

Tables 2 and 3 provides the age segregated subgroup analysis of AMH and AFC (Nomograms). The mean AMH and AFC values were higher in the fertile group compared to the infertile with the differences being significant in the age group of 20–25 and 25–30 years subgroups.

The centiles of AMH and AFC were computed at each age point using the linear specification of the CG-LMS method (Figs 2 and 3).

Age-related nomograms for AMH and AFC respectively were generated. Decline in AMH and AFC was linear in the infertile cohort while the fertile cohort had a non-linear pattern in decline. The rate of decline OF AMH was found to be 0.192 (± 0.05) in fertile group and 0.172 (± 0.04) in the infertile group. The rate of decline of AFC was found to be 0.65 (± 0.41) in fertile group and 0.56(± 0.05) in the infertile group. The estimated centiles for AMH and AFC for the fertile and infertile groups were generated (Figs 4A, 4B, 5A and 5B).

CG-LMS method using more complex models did not result in a significant improvement in the data-fit. Age cut off for decline in AMH and AFC in fertile women came out to be approximately 31 years using ROC analysis. Age cut off for decline in AMH and AFC in infertile women was found approximately 34 years. (Figs 6–9).

**Table 2. Nomograms for AMH using LMS model.**

| Age | 3-ctrl | 3-case | 10-ctrl | 10-case | 25-ctrl | 25-case | 50-ctrl | 50-case | 75-ctrl | 75-case | 90-ctrl | 90-case | 95-ctrl | 95-case |
|---|---|---|---|---|---|---|---|---|---|---|---|---|---|---|
| 20 | 2.22 | 0.83 | 3.25 | 1.91 | 4.37 | 3.28 | 5.7 | 5.01 | 7.1 | 6.89 | 8.41 | 8.67 | 9.22 | 9.77 |
| 21 | 1.93 | 0.73 | 3 | 1.75 | 4.18 | 3.09 | 5.59 | 4.8 | 7.08 | 6.65 | 8.49 | 8.42 | 9.35 | 9.51 |
| 22 | 1.65 | 0.63 | 2.74 | 1.6 | 3.98 | 2.9 | 5.46 | 4.59 | 7.04 | 6.43 | 8.54 | 8.19 | 9.47 | 9.28 |
| 23 | 1.37 | 0.54 | 2.47 | 1.44 | 3.75 | 2.71 | 5.31 | 4.38 | 6.98 | 6.22 | 8.56 | 7.99 | 9.55 | 9.09 |
| 24 | 1.11 | 0.45 | 2.2 | 1.28 | 3.52 | 2.52 | 5.13 | 4.18 | 6.89 | 6.04 | 8.55 | 7.82 | 9.59 | 8.94 |
| 25 | 0.88 | 0.38 | 1.94 | 1.14 | 3.27 | 2.34 | 4.94 | 3.99 | 6.77 | 5.86 | 8.51 | 7.67 | 9.6 | 8.8 |
| 26 | 0.7 | 0.31 | 1.69 | 1.01 | 3.02 | 2.17 | 4.73 | 3.81 | 6.62 | 5.69 | 8.43 | 7.52 | 9.56 | 8.67 |
| 27 | 0.55 | 0.27 | 1.47 | 0.9 | 2.77 | 2.01 | 4.5 | 3.63 | 6.44 | 5.52 | 8.31 | 7.38 | 9.48 | 8.55 |
| 28 | 0.43 | 0.23 | 1.26 | 0.8 | 2.53 | 1.86 | 4.27 | 3.47 | 6.23 | 5.37 | 8.15 | 7.25 | 9.35 | 8.43 |
| 29 | 0.34 | 0.2 | 1.09 | 0.71 | 2.3 | 1.73 | 4.02 | 3.31 | 5.99 | 5.23 | 7.94 | 7.13 | 9.16 | 8.34 |
| 30 | 0.27 | 0.17 | 0.93 | 0.64 | 2.08 | 1.6 | 3.76 | 3.17 | 5.72 | 5.09 | 7.67 | 7.01 | 8.9 | 8.24 |
| 31 | 0.22 | 0.15 | 0.8 | 0.58 | 1.86 | 1.49 | 3.49 | 3.02 | 5.42 | 4.93 | 7.35 | 6.87 | 8.58 | 8.1 |
| 32 | 0.18 | 0.13 | 0.68 | 0.52 | 1.67 | 1.39 | 3.22 | 2.87 | 5.11 | 4.75 | 7.01 | 6.68 | 8.22 | 7.91 |
| 33 | 0.15 | 0.12 | 0.59 | 0.48 | 1.5 | 1.28 | 2.99 | 2.72 | 4.82 | 4.56 | 6.69 | 6.47 | 7.88 | 7.69 |
| 34 | 0.13 | 0.11 | 0.52 | 0.43 | 1.37 | 1.19 | 2.79 | 2.57 | 4.59 | 4.38 | 6.42 | 6.26 | 7.6 | 7.47 |
| 35 | 0.11 | 0.09 | 0.47 | 0.39 | 1.26 | 1.1 | 2.64 | 2.43 | 4.4 | 4.21 | 6.22 | 6.07 | 7.4 | 7.27 |
| 36 | 0.1 | 0.09 | 0.42 | 0.36 | 1.17 | 1.02 | 2.5 | 2.3 | 4.24 | 4.04 | 6.05 | 5.88 | 7.22 | 7.08 |
| 37 | 0.09 | 0.08 | 0.39 | 0.33 | 1.09 | 0.95 | 2.37 | 2.18 | 4.08 | 3.88 | 5.86 | 5.7 | 7.03 | 6.89 |
| 38 | 0.08 | 0.07 | 0.35 | 0.3 | 1 | 0.88 | 2.23 | 2.04 | 3.88 | 3.69 | 5.63 | 5.47 | 6.77 | 6.65 |
| 39 | 0.07 | 0.06 | 0.31 | 0.26 | 0.92 | 0.79 | 2.07 | 1.86 | 3.64 | 3.41 | 5.31 | 5.1 | 6.41 | 6.22 |
| 40 | 0.06 | 0.05 | 0.28 | 0.22 | 0.82 | 0.65 | 1.87 | 1.57 | 3.33 | 2.9 | 4.89 | 4.36 | 5.92 | 5.34 |

**Table 3. Nomograms for AFC using LMS model.**

| Age | 3-ctrl | 3-case | 10-ctrl | 10-case | 25-ctrl | 25-case | 50-ctrl | 50-case | 75-ctrl | 75-case | 90-ctrl | 90-case | 95-ctrl | 95-case |
|---|---|---|---|---|---|---|---|---|---|---|---|---|---|---|
| 20 | 6.9 | 5.56 | 11.13 | 8.47 | 14.59 | 11.88 | 17.73 | 16.22 | 20.77 | 21.2 | 24.02 | 26.31 | 26.52 | 29.7 |
| 21 | 7.12 | 5.23 | 11.22 | 8.09 | 14.5 | 11.47 | 17.44 | 15.8 | 20.3 | 20.77 | 23.35 | 25.88 | 25.7 | 29.28 |
| 22 | 7.26 | 4.9 | 11.24 | 7.72 | 14.36 | 11.07 | 17.16 | 15.37 | 19.86 | 20.33 | 22.75 | 25.45 | 24.98 | 28.86 |
| 23 | 7.21 | 4.59 | 11.11 | 7.35 | 14.15 | 10.66 | 16.87 | 14.94 | 19.5 | 19.89 | 22.31 | 25.01 | 24.48 | 28.42 |
| 24 | 6.87 | 4.28 | 10.75 | 6.99 | 13.83 | 10.26 | 16.59 | 14.51 | 19.27 | 19.44 | 22.13 | 24.56 | 24.33 | 27.98 |
| 25 | 6.28 | 3.98 | 10.19 | 6.64 | 13.41 | 9.87 | 16.34 | 14.08 | 19.18 | 19 | 22.22 | 24.1 | 24.55 | 27.52 |
| 26 | 5.64 | 3.7 | 9.56 | 6.3 | 12.97 | 9.48 | 16.13 | 13.65 | 19.21 | 18.54 | 22.49 | 23.64 | 25.01 | 27.05 |
| 27 | 5.11 | 3.42 | 9 | 5.96 | 12.55 | 9.09 | 15.9 | 13.23 | 19.17 | 18.09 | 22.67 | 23.16 | 25.35 | 26.57 |
| 28 | 4.73 | 3.16 | 8.53 | 5.63 | 12.1 | 8.71 | 15.53 | 12.8 | 18.89 | 17.62 | 22.49 | 22.68 | 25.24 | 26.08 |
| 29 | 4.39 | 2.9 | 8.04 | 5.31 | 11.57 | 8.33 | 14.98 | 12.37 | 18.35 | 17.16 | 21.94 | 22.19 | 24.69 | 25.57 |
| 30 | 3.96 | 2.66 | 7.43 | 4.99 | 10.88 | 7.96 | 14.27 | 11.94 | 17.64 | 16.68 | 21.24 | 21.68 | 24 | 25.05 |
| 31 | 3.44 | 2.43 | 6.66 | 4.69 | 10.04 | 7.59 | 13.47 | 11.51 | 16.9 | 16.21 | 20.58 | 21.17 | 23.39 | 24.52 |
| 32 | 2.9 | 2.21 | 5.84 | 4.39 | 9.15 | 7.23 | 12.65 | 11.09 | 16.21 | 15.72 | 20.04 | 20.64 | 22.97 | 23.97 |
| 33 | 2.42 | 2.01 | 5.06 | 4.1 | 8.27 | 6.87 | 11.85 | 10.66 | 15.58 | 15.24 | 19.62 | 20.1 | 22.7 | 23.4 |
| 34 | 2.04 | 1.81 | 4.4 | 3.83 | 7.49 | 6.52 | 11.13 | 10.23 | 15.04 | 14.74 | 19.3 | 19.55 | 22.56 | 22.81 |
| 35 | 1.75 | 1.63 | 3.87 | 3.56 | 6.82 | 6.17 | 10.51 | 9.81 | 14.6 | 14.24 | 19.11 | 18.98 | 22.56 | 22.21 |
| 36 | 1.53 | 1.46 | 3.47 | 3.3 | 6.28 | 5.83 | 9.98 | 9.38 | 14.2 | 13.73 | 18.93 | 18.4 | 22.55 | 21.58 |
| 37 | 1.37 | 1.3 | 3.13 | 3.05 | 5.79 | 5.49 | 9.42 | 8.96 | 13.69 | 13.22 | 18.52 | 17.81 | 22.25 | 20.94 |
| 38 | 1.22 | 1.16 | 2.81 | 2.81 | 5.28 | 5.17 | 8.76 | 8.53 | 12.94 | 12.7 | 17.73 | 17.2 | 21.42 | 20.27 |
| 39 | 1.05 | 1.02 | 2.44 | 2.58 | 4.63 | 4.84 | 7.78 | 8.1 | 11.64 | 12.17 | 16.09 | 16.57 | 19.54 | 19.58 |
| 40 | 0.81 | 0.9 | 1.89 | 2.36 | 3.62 | 4.53 | 6.14 | 7.68 | 9.27 | 11.63 | 12.91 | 15.92 | 15.73 | 18.87 |

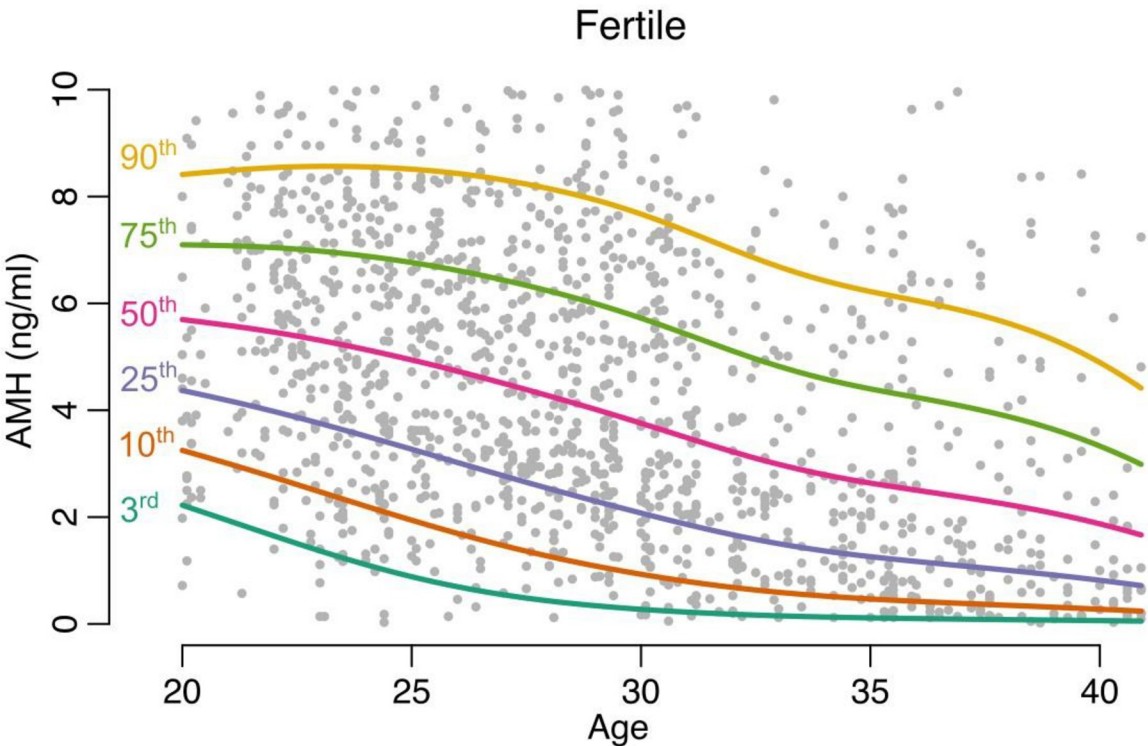

**Fig 2. Centiles of AMH computed at each age point (fertile population).**

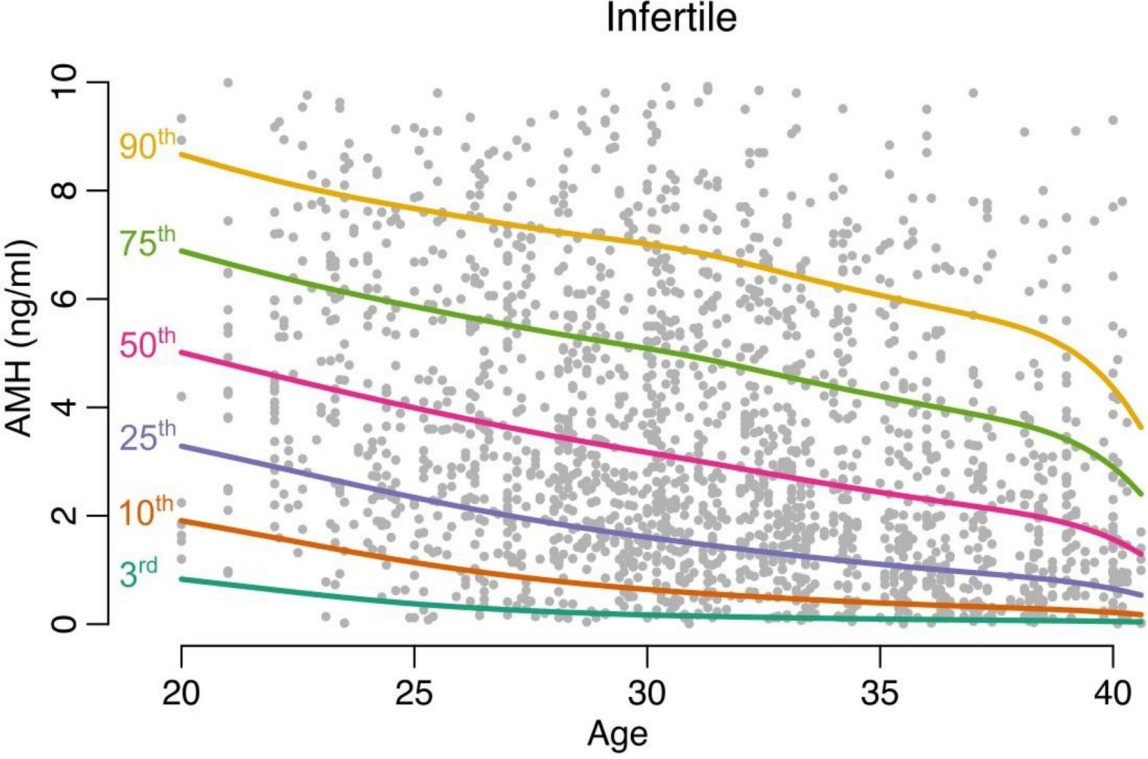

**Fig 3. Centiles of AMH computed at each age point (Infertile population).**

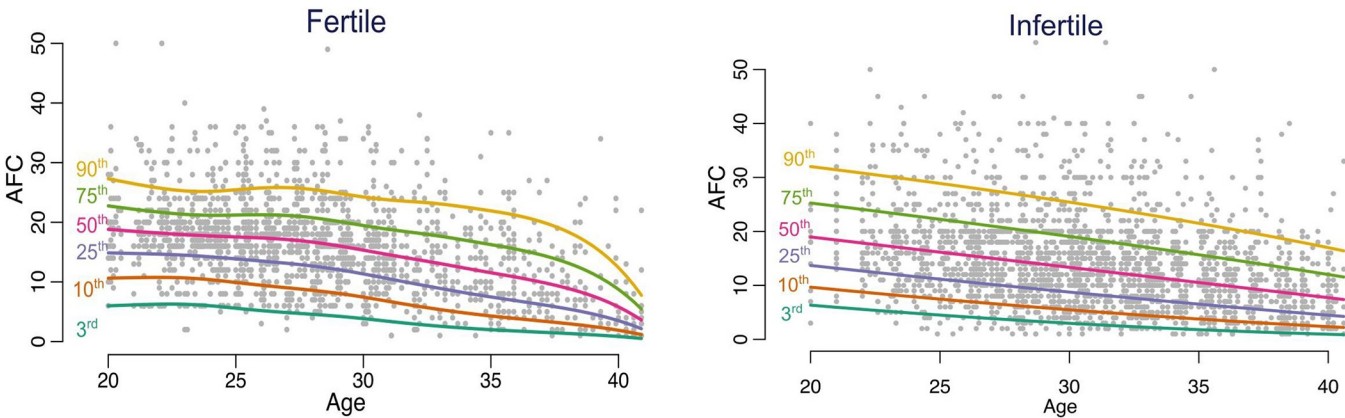

**Fig 4.** (a) Centiles of AFC computed at each age point (Fertile population). (b) Centiles of AFC computed at each age point (Infertile population).

The correlation between AMH and AFC was estimated for both infertile and fertile populations and it was found that both the parameters were significantly correlated (Fig 10)

## Discussion

The present study is the first of its kind to develop nomograms of Antimullerian Hormone and Antral follicle count for both fertile and infertile women of Indian origin. AMH and AFC are the two most frequently used ovarian reserve markers to prognosticate infertile couples regarding future fertility prospects. Our study with a large sample size showed the mean values of AMH to be significantly lower in infertile than fertile populations. As the patients with endometriosis, tuberculosis and PCOS were excluded, this difference in AMH values in

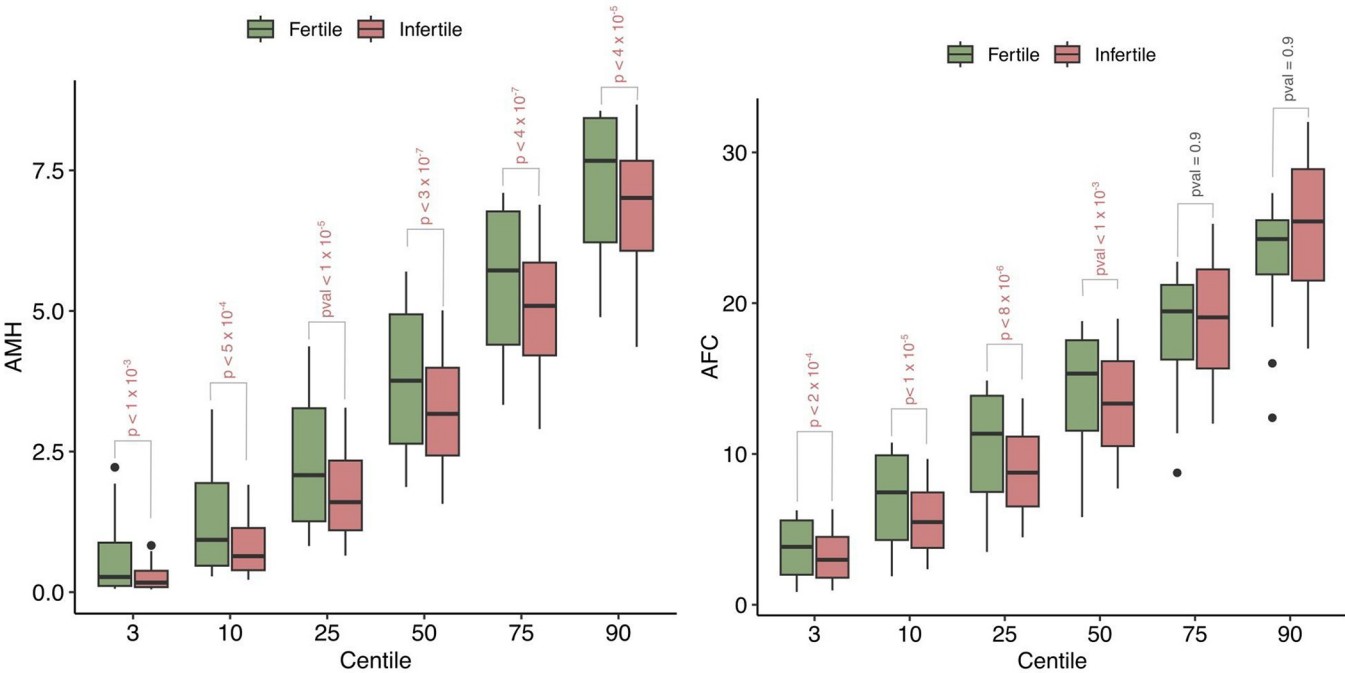

**Fig 5.** (a) The estimated centiles for AMH for the fertile and infertile groups. (b) The estimated centiles for AFC for the fertile and infertile groups.

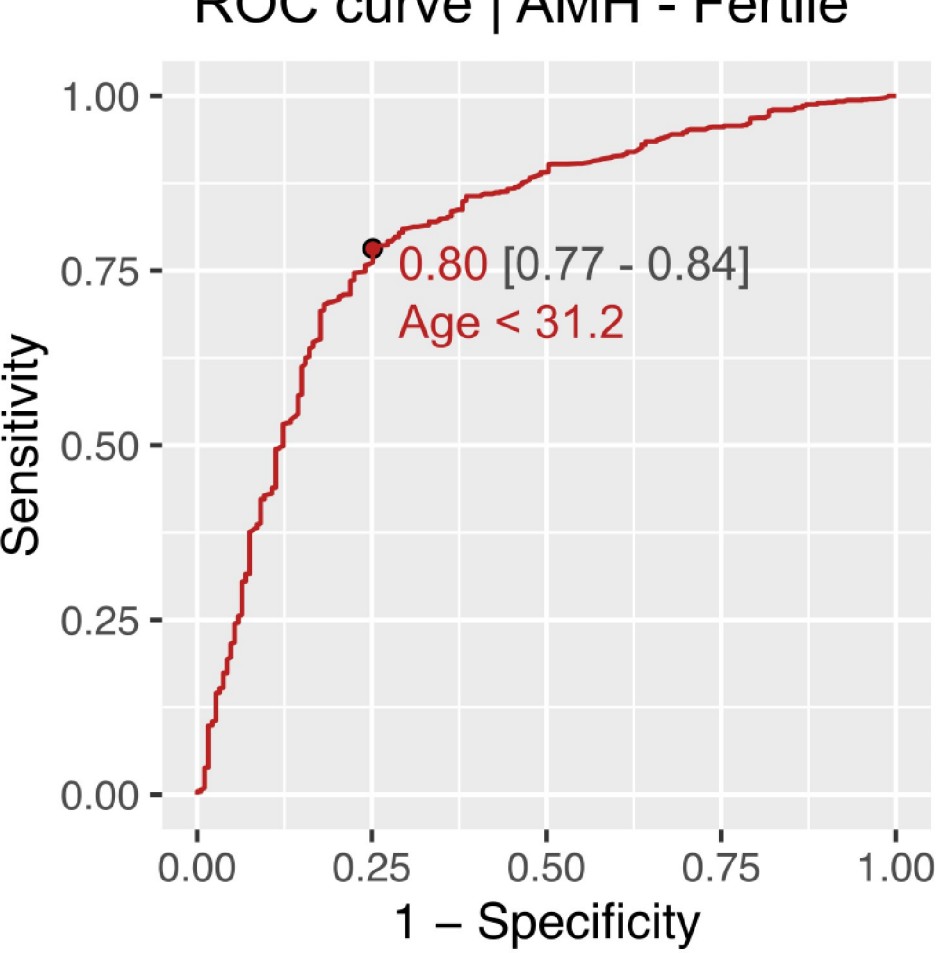

**Fig 6. Age cut off for decline in AMH in fertile women.**

corresponding age groups throws light on the possibility of certain ovarian factors at play causing decreased reserve in the infertile compared to age and BMI matched fertile women The difference in mean AMH values among infertile and fertile was seen across all centiles. Previously various authors have published nomograms and have shown similar age-related decline in AMH values [24, 25, 30]. Most studies showed a peak in AMH levels at 25 years with a further decline in the values. No such peak was seen in women from our study among the infertile group, though the fertile women showed a peak at higher centiles. This could be attributed to an ovarian insult at an early age, especially in the infertile group from a non-disease ovary, responsible for declining reserves. The age-related decline in AMH across all centiles was seen in a linear fashion in our study. However, studies by Nelson et al. (2011) and Seifer et al. (2011) among infertile Caucasian women showed that AMH declined in a non-linear pattern with age, which was best described by aquadratic equation; while the most recent study by Naasan et al. (2015) found that AMH declined in a linear biphasic fashion with age [31]. Nomograms for AFC were also computed for the infertile and fertile groups. The mean AFC was significantly higher in controls than cases across centiles though the difference diminished across higher centiles. A study by Iglesias et al where AMH and AFC were measured for the Indian and Spanish populations showed advancement in ovarian age in the Indian population

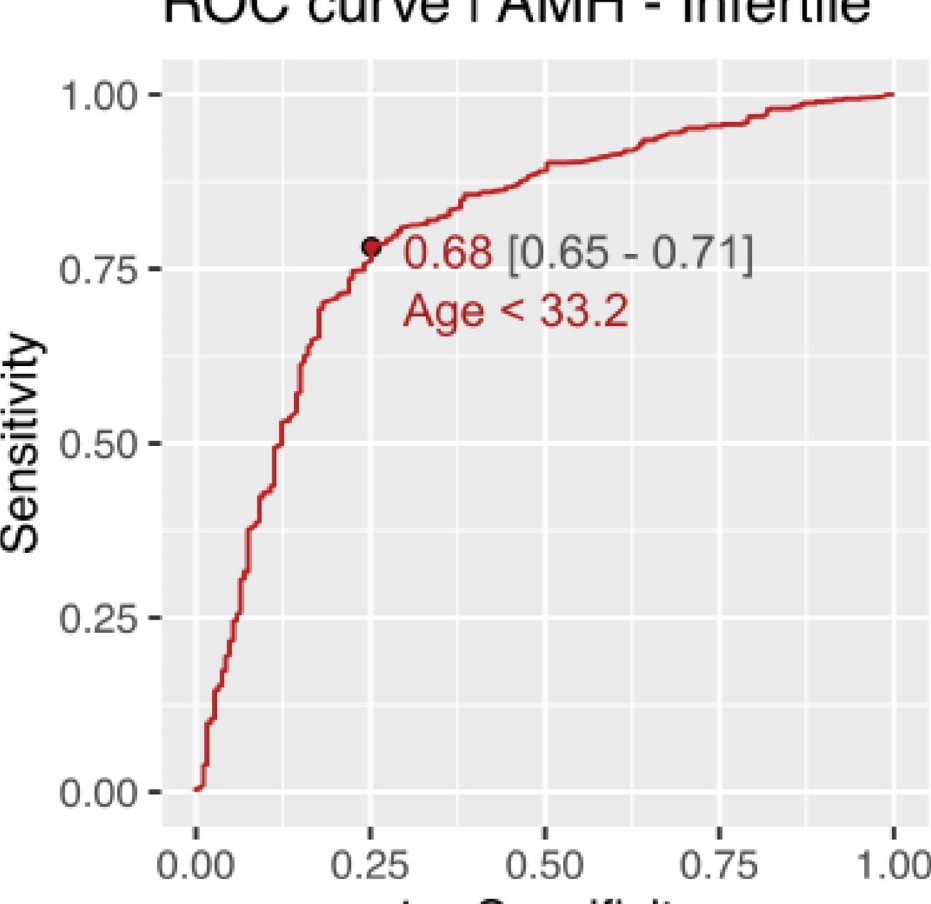

**Fig 7. Age cut off for decline in AMH in infertile women.**

compared to the Spanish population [27]. Our results contradict these findings as even the infertile population ovarian reserve markers were higher in our study than previously quoted. The findings of Olcha et al who studied the ancestry informative markers population' suggest that after controlling for age and BMI, AMH levels are not different based upon these genetic markers of ethnicity [32]. The age cut-off for decline in AMH (< 1.2 ngm/ml)was 31.2 and 33.2 years among the fertile and infertile women respectively, while it was 34.4 and 31.7 years among fertile and infertile women respectively for AFC (< 5). The cutoff was developed taking the POESIDON criteria to define diminished ovarian reserves [29]. This highlighted that women tend to decline ovarian reserves between 31.2 to 34.4 years given the above cut-off values for defining diminished ovarian reserves. This data may be useful in counseling women on birth planning, considering the trends on postponement in child bearing among Indian women for career pursuits.

There was positive correlation between AMH and AFC in our study. A study by Zhang et al demonstrated that one in five patients show discordance in AMH and AFC with AFC being better than AMH for DOR prediction [33]. As AFC reflect the recruited follicle and AMH an

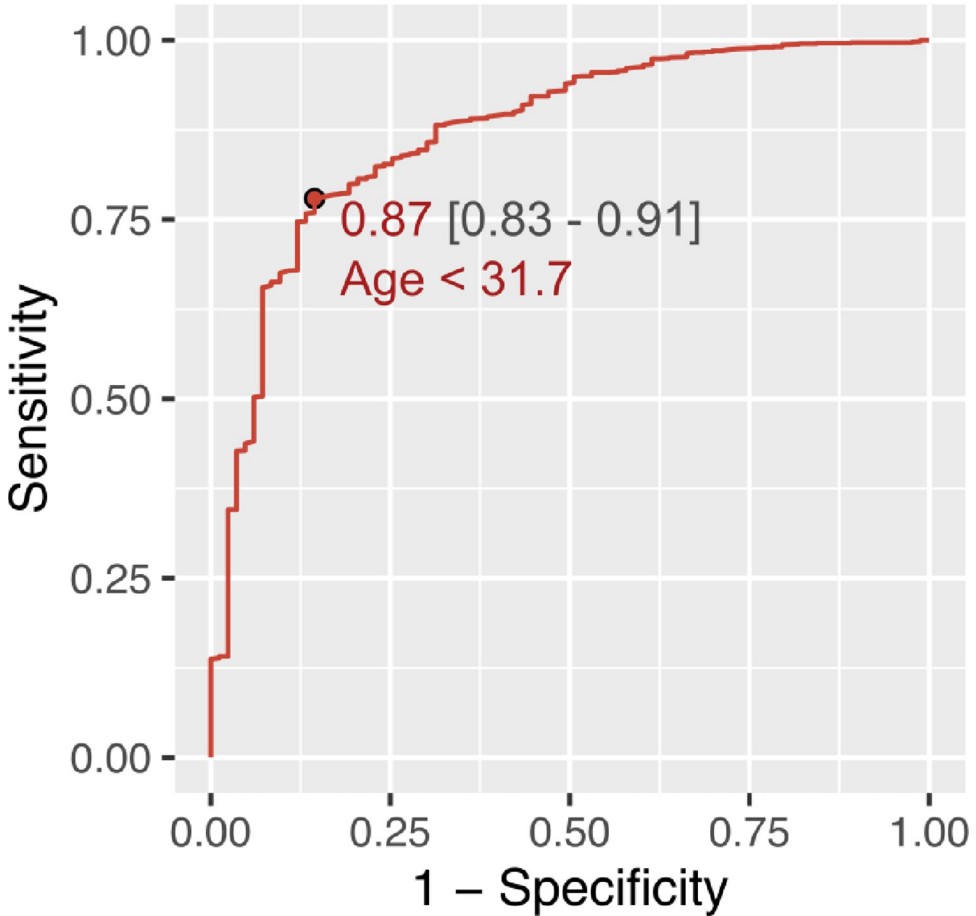

**Fig 8. Age cut off for decline in AFC in fertile women.**

ovarian reserve marker, AMH prognosticates the overall fertility potential further necessitating the need for nomograms.

When considering PCOS patients, stratifying patients based on AMH values can predict IVF outcomes in patients as shown by Reshef Tal et al where women with low AMHhad higher LBR compared to high AMH with PCOS [34]. The cut-off values to label a patient as PCOS depending on AMH has not been defined. The values >95th centiles of these nomograms can serve as an indicator to anticipate PCOS and plan further ART treatment. In a study conducted on chinese ethnic group in Singapore revealed that AFC and AMH generally decline in a linear fashion with increasing age, although there is a slight non-linear pattern observed for AMH in the 90th and 97th percentiles. These data showed that the ovarian reserve for older patients decreases at a constant rate over time. Faster decline rates of AFC and AMH were observed in higher centiles [30]

The strength of this study is the large sample size coming from a single center avoiding variations in the methods used to assess both the ovarian reserve markers, besides inclusion of adequate number of women from each age group. Further the assay used was Gen 2 ELISA, and all AFC measurements had little inter-observer variations. While the recent methods to assess

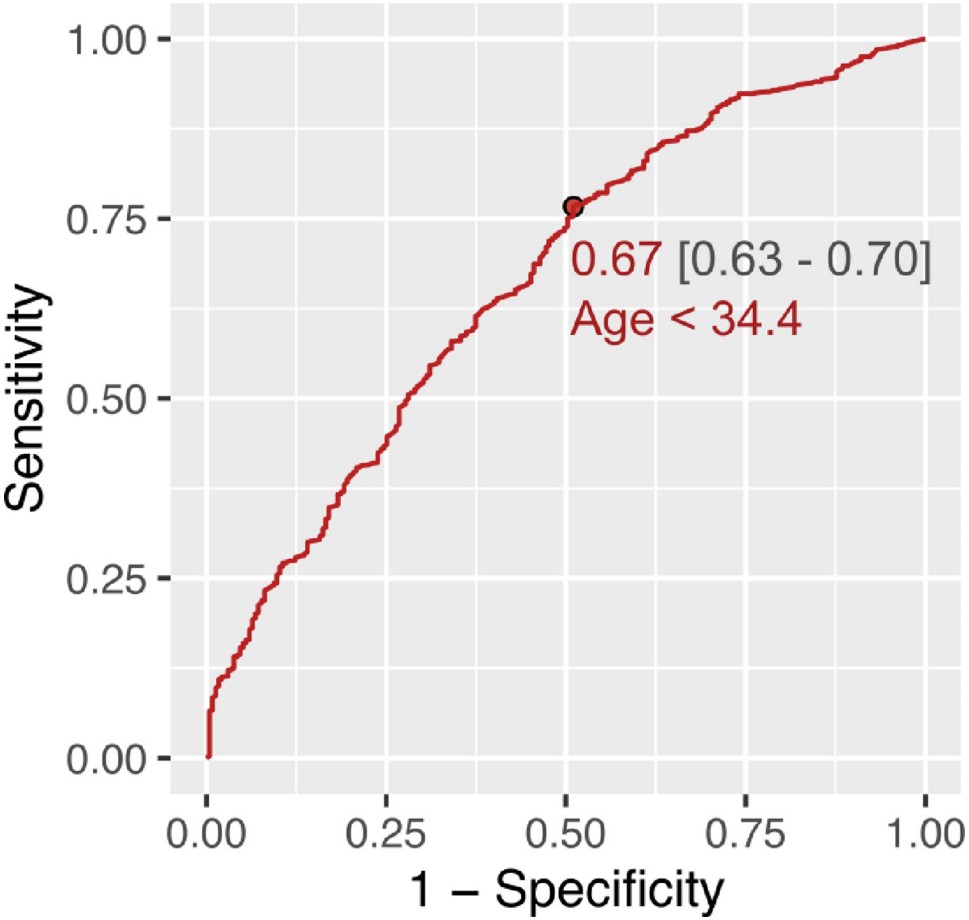

**Fig 9. Age cut off for decline in AFC in infertile women.**

AMH include automated assays, our study was initiated well before these assays were readily available in India. However, the difference between the ELISA Gen 2 and the automated assays can be computed from available studies and an always be factored while using these nomograms as reference by the clinicians.

The study had its limitation in particular, conducted on women from Northern India and variations among women hailing from other regions are anticipated given the large country with racial diversity. Further we did not have the outcomes from treatment cycles including ovarian stimulation with or without intra-uterine insemination and IVF, from the infertile cohort in terms of time to pregnancy, clinical pregnancy, live birth, eggs retrieved, fertilized and good quality embryos obtained and cryopreserved.

## Conclusion

The study provides nomograms to be used as reference levels for Indian females while assessing their ovarian reserve markers and reduce time to pregnancy by formulating an individualized plan. It gives insight on the age at which North Indian women suggest declining fertility and may be useful to advise on options available while women plan to defer child-bearing.

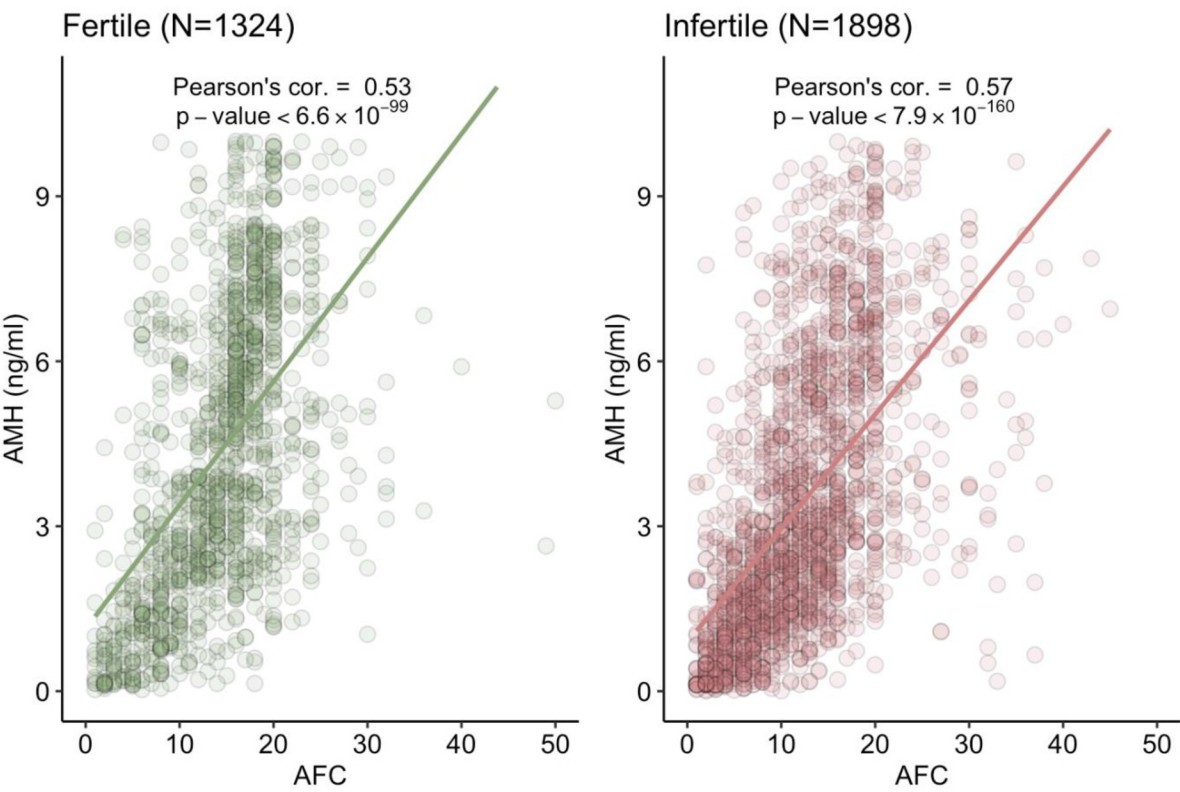

**Fig 10. Correlation between AMH and AFC for both infertile and fertile populations.**

## Supporting information

**S1 Rawdata.**
(XLSX)

## Author Contributions

**Conceptualization:** Neena Malhotra, Pankush Gupta, Pradeep Chaturvedi.

**Data curation:** Neena Malhotra, Pankush Gupta.

**Formal analysis:** Neena Malhotra.

**Funding acquisition:** Neena Malhotra.

**Investigation:** Neena Malhotra.

**Methodology:** Neena Malhotra, Pradeep Chaturvedi.

**Project administration:** Neena Malhotra.

**Resources:** Neena Malhotra.

**Software:** Neena Malhotra, Rintu Kutum.

**Supervision:** Neena Malhotra.

**Validation:** Neena Malhotra.

**Visualization:** Neena Malhotra.

**Writing – original draft:** Neena Malhotra, Pankush Gupta, Saloni Kamboj, Rintu Kutum.

**Writing – review & editing:** Neena Malhotra, Saloni Kamboj, Rintu Kutum.

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
