## [Decision Letter · Decision Letter 0]

1 Apr 2024

PONE-D-23-38873Age specific variations in ovarian reserves in healthy fertile and infertile women: A cross sectional studyPLOS ONE

Dear Dr. malhotra,

Thank you for submitting your manuscript to PLOS ONE. After careful consideration, we feel that it has merit but does not fully meet PLOS ONE’s publication criteria as it currently stands. Therefore, we invite you to submit a revised version of the manuscript that addresses the points raised during the review process.

The current manuscript may provide important information to the literature. After completion the required corrections of the reviewers, we will review again and make our final decision.  ==============================

We look forward to receiving your revised manuscript.

Kind regards,

Vehbi Yavuz Tokgöz, M. D.

Academic Editor

PLOS ONE

Journal Requirements:

3. Please amend your manuscript to include your abstract after the title page.

Reviewers' comments:

Reviewer's Responses to Questions

**Comments to the Author**

1. Is the manuscript technically sound, and do the data support the conclusions?

Reviewer #1: Yes

2. Has the statistical analysis been performed appropriately and rigorously? 

Reviewer #1: Yes

3. Have the authors made all data underlying the findings in their manuscript fully available?

Reviewer #1: Yes

4. Is the manuscript presented in an intelligible fashion and written in standard English?

Reviewer #1: Yes

5. Review Comments to the Author

Reviewer #1: This is a cross sectional study and provide us nomograms for AMH and AFC among fertile and infertile women of Indian origin, indicating the trends in decline and the relationship between two ovarian reserve markers. The present study aims to derive nomograms for AMH and AFC among fertile and infertile women of Indian origin, indicating the trends in decline and the relationship between two ovarian reserve markers. The researchers included 1902 infertile and 1338 healthy women for controls. This is a cross sectional study and has a good sample size.

The following points should be corrected and clarified:

- The authors should change the sentence 'Assessment of AFC has been a simple method of predicting the occurrence of menopause and thus the duration of the reproductive lifespan'. Because we know, based on the work of Kim C. et al. ‘Among women aged 45-49 years, undetectable AMH concentrations were associated with a greater than 60% probability of menopause within 5 years, whereas approximately 1/3 of women with no or just one antral follicle experienced menopause within 5 years. Both low and high concentrations of FSH were associated with greater odds of menopause than intermediate concentrations. Models with multiple markers did not improve the prediction of menopause over that afforded by models with single markers’. (Maturitas 2017 Aug:102:18-25).

- In the infertile population, were couples with infertility due to male factor and tubal factor excluded?

- The p value acronyms in the figures should always be arranged in the same format

- Explanations on figures and tables should be clearer

- Spelling and typographical errors in the article should be corrected

6. PLOS authors have the option to publish the peer review history of their article (what does this mean?). If published, this will include your full peer review and any attached files.

Reviewer #1: No

---

## [Author Response · Author response to Decision Letter 0]

5 May 2024

RESPONSE TO EDITORS 

1. Formatting done as per Journal protocol

2. Raw Data Attached

3. Abstract included after Title page

4. Corrected Reference list attached

RESPONSE TO REVIEWERS

The authors should change the sentence 'Assessment of AFC has been a simple method of predicting the occurrence of menopause and thus the duration of the reproductive lifespan'. Because we know, based on the work of Kim C. et al. ‘Among women aged 45-49 years, undetectable AMH concentrations were associated with a greater than 60% probability of menopause within 5 years, whereas approximately 1/3 of women with no or just one antral follicle experienced menopause within 5 years. Both low and high concentrations of FSH were associated with greater odds of menopause than intermediate concentrations. Models with multiple markers did not improve the prediction of menopause over that afforded by models with single markers’. (Maturitas 2017 Aug:102:18-25).

ANSWER : editing done ( AMH and FSH also added for predicting menopause ).

- In the infertile population, were couples with infertility due to male factor and tubal factor excluded?

ANSWER : Severe Male factor infertility were excluded but mild male factor were included. Tubal factor infertility were all included.

- The p value acronyms in the figures should always be arranged in the same format

Answer : Changed as per requirements

- Explanations on figures and tables should be clearer

Answer : Captions added. 

- Spelling and typographical errors in the article should be corrected

Answer : Corrections done.

---

## [Editor Report · Decision Letter 1]

1 Aug 2024

Age specific variations in ovarian reserves in healthy fertile and infertile women: A cross sectional study

PONE-D-23-38873R1

Dear Dr. Nenna Malhotra

We’re pleased to inform you that your manuscript has been judged scientifically suitable for publication and will be formally accepted for publication once it meets all outstanding technical requirements.

Kind regards,

Natasha L Pritchard

Academic Editor

PLOS ONE

Additional Editor Comments (optional):

Dear author

Your responses are appropriate. I think that it should be clearly stated that tubal factor/ mild male factor infertility is included in the infertile group. This would only bias your results towards the null hypothesis regardless.

Thank you for your submission.
---

## [Editor Report · Acceptance letter]

22 Aug 2024

PONE-D-23-38873R1 

PLOS ONE

Dear Dr. malhotra, 

I'm pleased to inform you that your manuscript has been deemed suitable for publication in PLOS ONE. Congratulations! Your manuscript is now being handed over to our production team.

Kind regards, 

on behalf of

Dr. Natasha L Pritchard 

Academic Editor

PLOS ONE